# Food Waste (Beetroot and Apple Pomace) as Sorbent for Lead from Aqueous Solutions—Alternative to Landfill Disposal

Tatjana Šoštarić [1,*], Marija Simić [1], Zorica Lopičić [1], Snežana Zlatanović [2], Ferenc Pastor [3], Anja Antanasković [1] and Stanislava Gorjanović [2]

1 Institute for Technology of Nuclear and Other Mineral Raw Materials (ITNMS), Franchet d'Eperey 86, 11000 Belgrade, Serbia
2 Institute of General and Physical Chemistry, University of Belgrade, Studenstski trg 12/V, 11000 Belgrade, Serbia
3 Faculty of Chemistry, University of Belgrade, Studentski trg 12-16, 11000 Belgrade, Serbia
* Correspondence: t.sostaric@itnms.ac.rs

**Abstract:** This article presents studies, whose main goal was to minimize food waste. To achieve this goal, it is necessary to expand the scope of their application, for example, for the purification of polluted water from heavy metals. Millions of tons of waste from the fruit and vegetable industry, including pomace of apples and beetroots, are thrown into landfills, posing a danger to the environment. In order to solve the problems with the disposal of these wastes, the authors investigated their sorption potential for the removal of lead from wastewater. The sorbents, dried apple (AP), and beetroots (BR) pomaces were characterized by various methods (study of composition, zeta potential, FTIR-ATR, and SEM-EDX). Various models of sorption kinetics and sorption isotherms were analyzed. Kinetical studies under optimal conditions showed that the sorption process occurs through complexation and ion exchange and the determining stage limiting the rate of sorption is the diffusion of lead ions in the sorbent. The maximum sorption capacity was 31.7 and 79.8 mg/g for AP and BR, respectively. The thermodynamic data revealed the spontaneous sorption of lead ions by sorbents. The temperature rise contributes to the sorption increase by the AP sorbent, while for the BR sorbent, the opposite effect is observed. The obtained results showed that apple and beetroots pomaces can serve as effective renewable materials for the preparation of sorbents, contributing to the solution of complex environmental problems.

**Keywords:** organic waste; apple pomace; beetroot pomace; lead sorption





## 1. Introduction

An increase in global food production/consumption inevitably leads to increased waste generation. Food waste induces a loss of resources and has a harmful environmental impact that covers all emissions derived throughout the entire supply chain. Food loss and food waste occur at different stages of the supply chain and are caused by different reasons [1]. Both depend on the product and region: in middle/high-income countries, it mainly occurs during the distribution and consumption phases, while in low-income countries, the same thing happens during the production and post-harvest phases. The later the product is wasted in the supply chain, the greater its environmental impact, due to the cumulative effect of all emissions during production, processing, transportation etc. [2].

According to the FAO, in addition to roots and tubers, fruits and vegetables have the highest wastage rates of any food products. Almost half of all the fruit and vegetables produced are wasted. In Serbia, food processing and food waste management are extremely limited. According to a report by the German Agency for International Cooperation titled "Climate Sensitive Waste Management Project in Serbia" [3], 770,000 tons of food are wasted annually and only less than 10% is recycled. The report estimates that about 90% of the total waste is still disposed of at non-sanitary municipal landfills or even worse, at illegal

dumpsites. The report also suggests that providing another route for food waste before it reaches the landfills can results in savings of 580 kg $CO_2$ eq per ton.

According to the FAO, world fruit production has increased by 54% in the last 20 years, reaching 883 million tons, an increase of 311 million tons. Five fruit species, namely bananas and plantains (18%), watermelons (11%), apples (10%), oranges (9%) and grapes (9%), accounted for 57% of the total production [4]. In the last two years, global apple production was about 82 million tons, with Serbia being one of the largest apple producers in South-East Europe, with an average production of 468,000 tons over the last five years [5]. Processed apples are usually used for making beverages, and the main by-product of the processing is apple pomace, which represents 10–35% of fresh fruit. This perishable waste contains mainly skin, pulp (70–76%), seeds (2.2–3.3%), and stalks (0.4–0.9%) and is susceptible to fast fermentation due to its high moisture content, which causes serious problems [6]. The composition of apple pomace depends of the apple variety and processing technology used [7]. According to Bhushan et al. [8], dried apple pomace contains 3.9–10.8% moisture, 0.5–6.1% ash, 2.94–5.67% proteins, 48–62% total carbohydrates, 36.5% insoluble fibers, 14.6% soluble fibers, 1.2–3.9% fat and 3.5–14.32% pectin.

In comparison to apples, beetroot (*Beta vulgaris* L.) production and consumption are currently on a smaller scale. However, beetroot is becoming more attractive and accepted by the consumers due to the high levels of desirable compounds, such as folate, manganese, potassium, iron, vitamin C, etc., that promote human health. Since juices consumption has faced a decade-long decline in the US and Europe due to the high levels of sugar content, the still and soft drink industry had to shift towards low and no-calorie products by using vegetables such as beetroot to reduce overall sugar content [9]. After squeezing the beetroot, pomace (15–30%) is often considered as a waste. Beetroot pomace contains significant amounts of fiber, as well as high concentrations of phenolic compounds and nitrogenous pigments called betalains. However, beetroot pomace is usually sent to landfills or rarely used as livestock feed [10]. According to Costa et al. [11], beetroot pomace contains 10.1% moisture, 5.62% ash, 12.64% proteins, 20.83% total carbohydrates, 45.08% insoluble fiber, 20.14% soluble fiber and 1.31% fat.

By relaying on Landfill Directive (1999/31/EC) (Council Directive 1999/31/EC, 1999), EU Waste Framework Directive (2008/98/EC) and the Packaging and Packaging Waste Directive (94/62/EC) (Directive, E.C., 1994) the European Commission pronounces a new targets to reduce landfilling by 2025 for recyclable waste (including bio-waste) in landfills, corresponding to maximum landfilling rate of 25% [12]. In order to diminish amounts of food waste, the emission of greenhouse gases, and provide a new value of these materials, sustainable solutions needs to be applied [13].

A number of studies have been reported for heavy metal ions removal by using different kind of food waste, such as sunflower seed husks [14], Oleaster seed [15], peach stone [16], banana peel [17], pecan nutshell [18], corn silk [19], *Solanum melongena* leaf [20], apple pomace [21], apples, pears, chokeberry and rosehip pomace [22], etc.

The study proposed in this paper aims to investigate the potential of apple and beetroot pomace as low-cost and effective sorbents for removing lead ions from synthetic solutions. The chemical composition of pomace makes it suitable for heavy metal binding due to the presence of functional groups that have an affinity towards cations (–CO, –COO, –OH and –$NH_2$) [23]. The study examines the impact of different operating parameters, such as pH, temperature, contact time, initial lead concentration, and sorbent dosage, on the capacity of the sorbents to remove lead ions from the solution. The data obtained will be simulated using different isotherm and kinetic models to evaluate the lead sorption onto the pomace and determine the maximum capacity and nature of the sorption mechanism. This study has the potential to provide a sustainable solution for the utilization of pomace waste and contribute to the reduction of heavy metal pollution in the environment.

## 2. Materials and Methods

### 2.1. Biosorbent Preparation

The apple pomace was obtained from plant "Fruvita" (Smederevo, Serbia) and the beetroot pomace was obtained from plant "Zdravo" (Selenča, Serbia), at industrial scale level. No treatment other than squeezing of thoroughly washed apples and beetroots was used. Both pomaces were collected aseptically, after squeezing and dehydrated as described in patent [24]. Dehydrated pomace were ground to a particle size below 300 microns. Obtained samples were marked as AP (apple) and BR (beetroot).

### 2.2. Metal Solution Preparation

Lead stock solution (1000 mg/L) was prepared by dissolving required mass of $Pb(NO_3)_2 \cdot 3H_2O$ (analytical grade) in deionized water. For the experimental purposes, the initial solution was further diluted with deionized water to obtain desired concentrations.

### 2.3. Characterization of AP and BP

LabSwift-aw, Novasina AG, Switzerland AW-meter was used to measure the water activity of the samples at 25 °C [24].

In order to determine mineral composition, materials were dissolved by using standardized microwave-assisted acid dissolution procedure (EPA Methods 3052) in High-performance Microwave Digestion System ETHOS UP, Milestone. After sample digestion, minerals concentrations were detected by atomic absorption spectrometry (AAS) using PerkinElmer PinAAcle 900T, USA. Analyses were performed with three replicates, where the average values are presented.

Scanning Electron Microscopy—Energy Dispersive X-ray Spectroscopy (SEM-EDX) analysis was performed with dried samples coated under vacuum with thin layer of gold and observed using a JEOL JSM-6610 LV model (JEOL Ltd., Japan).

Cation exchange capacity (CEC) of the both samples was determined by soaking 0.2 g of the sample in 100 mL of 1.0 mol/L ammonium acetate [14]. After 120 min of shaking at 250 rpm, the suspension was filtered and supernatant with released exchangeable cations ($K^+$, $Na^+$, $Ca^{2+}$ and $Mg^{2+}$) has been analyzed on AAS.

In order to determine the surface charge of the samples, zeta potential was measured by using Zetasizer Nano Z (Malvern, UK). The samples were dispersed in distilled water which pH value was adjusted at pH 5.0. The measurements of zeta potential were repeated five times, and the average values are presented. Refractive index (RI) for both samples was 1.35.

Surface functional groups were determined by Fourier Transform Infrared Spectroscopy (FTIR-ATR mode) using Thermo Nicolet 6700 FTIR (International Equipment Trading Ltd., USA). Both samples were analyzed before and after the sorption of lead, in order to record changes induced by the metal binding, as well as material behavior during the sorption process.

### 2.4. Sorption Experiments

In order to investigate effect of different operating parameters onto AP and BR sorption behavior, sorption experiments were performed in a batch system. The suspensions were agitated on orbital shaker (Heidolph, Unimax, Germany) at 250 rpm. After filtration (Whatman 542) lead residual concentration in solution was determined by AAS. The amount of sorbed $Pb^{2+}$ onto sorbent ($q_e$ (mg/g)) is calculated using the following equation:

$$q_e = \frac{(C_i - C_e) \times V}{m} \tag{1}$$

where $V$ (mL) is volume of the solution, $m$ (g) is a weight of the sorbent, $C_i$ (mg/L) and $C_e$ (mg/L) are the initial and the equilibrium $Pb^{2+}$ concentration in solution, respectively. All experiments were conducted in triplicate, and the average value is presented.

Effect of the initial pH value of lead solution was studied in the range between pH 2.0 and 6.0 by adding 0.1 mol/L $HNO_3$ or 0.1 mol/L KOH. In order to prevent metal hydroxide precipitation pH values higher than 6 weren't included in investigations. Initial lead concentration in solution was 200 mg/L. The mixture was shaken on orbital shaker at 250 rpm for 120 min. The solid/liquid ratio was 2 g/L. After filtration the residual metal concentration was determined by using AAS. The initial and the final pH values of the solution were measured by pH meter (Hach Sension+ MM340 GLP). Effect of sorbent concentration was performed by varying concentrations from 2 to 12 g/L. In this set of experiments initial metal concentration was 200 mg/L, while initial pH of solution was set to 5.0. Contact time between solid and liquid phase was 120 min. Effect of contact time on sorption capacity, was investigated in range from 2 to 180 min. Effect of initial lead concentration was studied in range from 40 to 400 mg/L, while all other parameters were the same as it was described previously. Investigations of the temperature effects on $Pb^{2+}$ sorption onto both sorbents were conducted in the range from 293 K to 323 K, under the following operational conditions: $C_i$ = 200 mg/L, pH = 5, solid/liquid ratio 2 g/L, contact time 120 min and stirring speed 250 rpm. Obtained experimental data were used in thermodynamic studies.

### 2.5. Kinetic and Isotherm Studies

In order to get more profound explanation of bonding mechanism, experimental data were fitted by different kinetic and isotherm models listed in Table S1.

### 2.6. Thermodynamic Study

Sorption of divalent cations onto the solid sorbent surface can be described as reversible heterogeneous sorption process [25]. In order to conclude whether this process is spontaneous or not, it's thermodynamic consideration is necessary to be done. An indicator of chemical reaction spontaneity is the Gibb's free energy change, $\Delta G^0$, including both changes in enthalpy ($\Delta H^0$) and entropy ($\Delta S^0$) of the sorption process. The sorption process free energy is related to the equilibrium constant, and can be described by the following equation:

$$\Delta G^0 = -RTlnK_e^0 \tag{2}$$

where $R$ represents universal gas constant (8.314 J/mol/K) and $T$ is temperature in K, while $K_e^0$ (dimensionless) represents the thermodynamic equilibrium constant of adsorption.

After rearranging, Equation (2) becomes:

$$K_e^0 = \exp\left(\frac{-\Delta G^0}{RT}\right) \tag{3}$$

Constant $K_e^0$ might be calculated from distribution constant ($K_d$), which represents distribution of metal ions between solid and liquid phase after equilibrium, defined as:

$$K_d = \frac{q_e}{C_e} \tag{4}$$

where $C_e$ (mg/L) is the equilibrium concentration and $q_e$ (mg/g) is the sorption capacity at equilibrium.

In order to transform $K_d$ to dimensionless $K_e^0$, it is necessary to multiply this constant by 1000 [26,27].

The fundamental thermodynamic relation of three thermodynamic parameters ($\Delta G^0$, $\Delta H^0$ and $\Delta S^0$) is described as:

$$\Delta G^0 = \Delta H^0 - T\Delta S^0 \tag{5}$$

Assuming that the changes in $\Delta H^0$ and $\Delta S^0$ with temperatures are negligible, after substituting Equation (3) into Equation (5), the linear (Equation (6)) form of the well-known van't Hoff equation is achieved.

$$lnK_e^0 = -\Delta H^0/R + \Delta S^0/RT \tag{6}$$

In this paper, thermodynamic study was conducted in temperature range from 293 K to 323 K under operational conditions previously determined. Parameters $\Delta H^0$ and $\Delta S^0$ are determined from the slope and the intercept of the plot of ln $K_e^0$ vs. $1/T$, and were used, along with standard Gibbs free energy change ($\Delta G^\circ$), to explain the sorption behaviour.

## 3. Results and Discussion

### 3.1. Characterisation of Sorbents

Previously, underutilized by-product AP was transformed into stable apple pomace flour by process of dehydration and afterward it was comprehensively characterized [28–30]. Organic composition of AP and BR was determined by Zlatanović et al. [29] and Jovanović et al. [31], respectively, while inorganics composition of both materials was determined and the results are presented in Table 1. According to Zlatanović et al. [29] moisture content of AP is between 4.0 to 7.7%, ash content is between 1.2 to 1.9% while water activity (aw) is between 0.2 and 0.4 [29]. AP and BP used within the scope of this study were obtained in the same manner as reported [24,32].

**Table 1.** Chemical composition of AP and BR.

| Composition | AP | BR |
|---|---|---|
| Fat (g/100 g dw) | $1.3 \pm 0.20$ | $0.92 \pm 0.12$ |
| Proteins (g/100 g dw) | $3.2 \pm 0.30$ | $13.85 \pm 0.34$ |
| Total carbohydrates (g/100 g dw) | $50 \pm 6.00$ | $34.58 \pm 0.23$ |
| Total content of fiber (g/100 g dw) | $45.0 \pm 4.00$ | $26.7 \pm 3.30$ |
| Cellulose (g/100 g dw) | $18.0 \pm 1.00$ | $9.3 \pm 0.30$ |
| Soluble fiber (g/100 g dw) | $3.0 \pm 0.20$ | $7.1 \pm 0.30$ |
| Inorganic composition (%) | | |
| Cu | $<0.00005 \pm 0.00$ | $<0.00005 \pm 0.00$ |
| Zn | $0.0021 \pm 0.00$ | $0.0041 \pm 0.00$ |
| Ni | $0.0020 \pm 0.00$ | $0.0015 \pm 0.00$ |
| Pb | $0.0005 \pm 0.00$ | $0.0055 \pm 0.00$ |
| Fe | $0.014 \pm 0.00$ | $0.011 \pm 0.00$ |
| Mn | $0.0005 \pm 0.00$ | $0.0012 \pm 0.00$ |
| Cd | $0.0006 \pm 0.00$ | $0.0006 \pm 0.00$ |
| Mg | $0.067 \pm 0.04$ | $0.130 \pm 0.06$ |
| K | $0.550 \pm 0.01$ | $2.400 \pm 0.08$ |
| Na | $0.0058 \pm 0.00$ | $0.140 \pm 0.06$ |
| Ca | $0.090 \pm 0.02$ | $0.204 \pm 0.08$ |

Due to the evident presence of exchangeable ions (potassium, calcium, sodium and magnesium) which often take part in ion-exchange mechanism, it can be concluded that both samples have a potential to be a good metal sorbents. In comparison to AP, BR has significantly higher concentration of all exchangeable cations.

### 3.2. SEM-EDX

In order to observe the samples surface morphology SEM-EDX analyses have been performed (Figure 1). As can be seen the BR sample has a layered fibrous morphology with noticeable gaps between fibers and pores inside them. Since the AP consist of different types of materials like skin, flesh and apple seeds, the sample structure is not consistent and the micrographs vary depending on fruit part that is shown. Micrographs presented at Figure 1a,b shows apple flash which is more porous than skin and seeds, due to hemicellu-

lose structure [33]. Surfaces, like these, with a larger contact area and more binding sites for ions, are more accessible to metal ions.

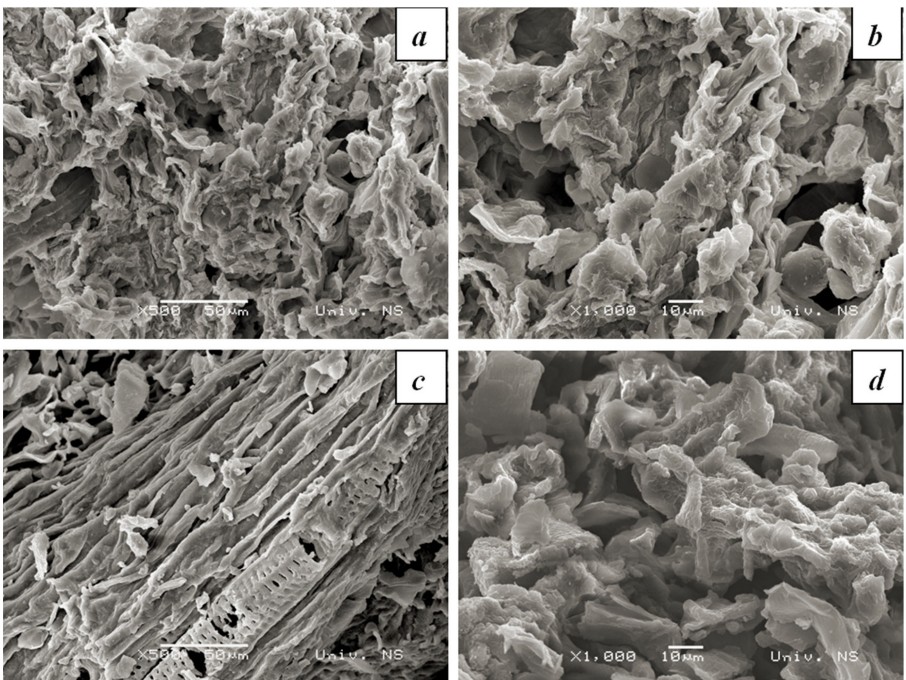

**Figure 1.** SEM micrographs of AP at magnification of 500× (**a**) and 1000× (**b**) and BR at magnification of 500× (**c**) and 1000× (**d**).

Figure 2a,b, shows EDX spectrum of AP and BR, respectively. In comparison to the EDX spectrum of AP, EDX spectrum of BR revealed the presence of alkali (potassium, sodium) and alkaline earth ions (magnesium, calcium) on surface which is accordance with Table 1, where is evident that BR have much more of these ions.

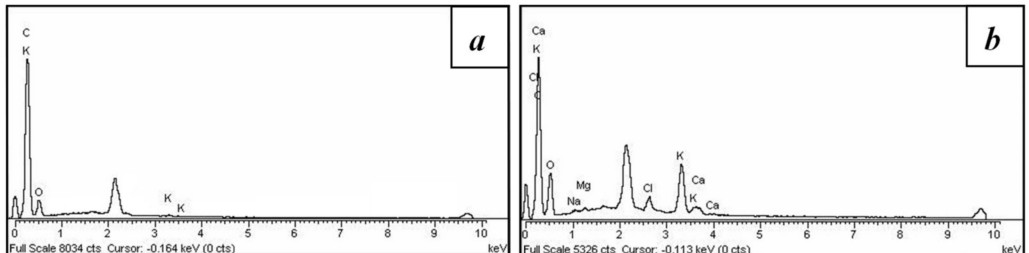

**Figure 2.** EDX analysis of AP (**a**) and BR (**b**).

*3.3. Cation Exchange Capacity—CEC*

Cation exchange capacity (CEC) was calculated and the results are shown at Table 2. The results are presented in miliequivalents per 100 g of sorbents (meq/100 g).

**Table 2.** Cation exchange capacity (CEC) of AP and BR.

|  | $Na^+$ | $K^+$ | $Ca^{2+}$ | $Mg^{2+}$ | **CEC** |
|---|---|---|---|---|---|
|  | meq/100 g | | | | |
| **AP** | 0.65 | 7.80 | 11.2 | 5.10 | 24.75 |
| **BR** | 5.40 | 72.95 | 9.00 | 8.50 | 95.85 |

As can be seen from Table 2, CEC is 24.75 and 95.85 meq/100 g for AP and BR, respectively. BR has significantly higher cation exchange capacity, due to potassium ions

(confirmed in Table 1) which are dominant ions in the exchangeable positions. The result indicates that ion exchange mechanism can have an important role during the process of lead sorption.

### 3.4. Effect of Operation Parameters on Sorption Process

Since many factors, such as initial sorbate pH, contact time, ratio between solid and liquid phase, initial lead concentration, temperature etc. can affect the effectiveness of sorption process, in this section results of this investigations are presented (Figure 3).

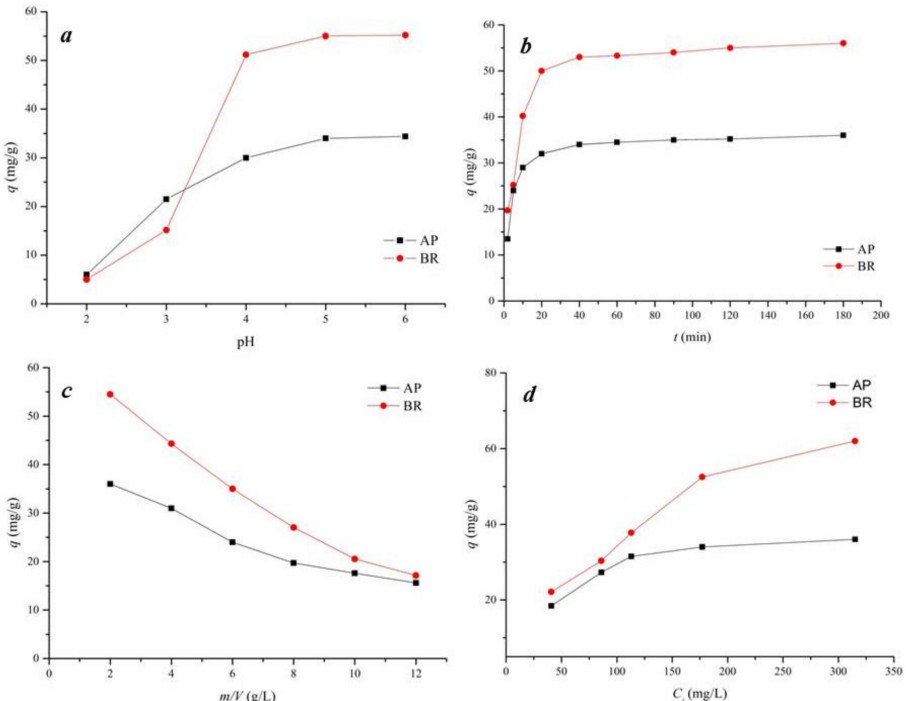

**Figure 3.** Effect of different operation parameters on sorption capacity: (**a**) pH, (**b**) contact time, (**c**) sorbents concentration and (**d**) initial $Pb^{2+}$ concentration (constant parameters were pH 5.0 (**b–d**), contact time 120 min (**a,c,d**), sorbent concentration 2 g/L (**a,b,d**) and initial $Pb^{2+}$ concentration 200 mg/L (**a–c**).

From Figure 3a it is evident that the sorption capacity (q) increases with increasing pH up to 5.0. Apparently BR has considerably higher sorption capacity in comparison to AP. Both biosorbents have higher affinity toward $Pb^{2+}$ at higher pH values due to protonation of surface functional groups at lower pH values. It is well known that metal ions compete with hydrogen ions. At higher pH value of solution, surface deprotonation occurs causing better accessibility to positively charged lead ions, which often results in higher sorption capacity of sorbent [34]. Later in the text zeta potential analysis also revealed that both sorbents are negatively charged (at pH 5.0 values of zeta potential are −36.8 and −40.1 mV for AP and BR, respectively). All this implies the presence of negatively charged functional groups on sorbents surface, indicating that materials with such surface charges are capable of attracting positively charged metal cations [35]. Since, the maximum sorption capacity of both sorbents took place at pH value 5.0 all further experiments were performed at initial pH value of 5.0.

The results of contact time are presented at Figure 3b. It can be observed that process of lead sorption onto both sorbents occurs in three phases: initial rapid phase within the first minutes of contact, second slower phase from 5 to 30 min, after which the sorption capacity becomes nearly constant (third phase). Due to high initial metal concentration in solution together with numerous available sorption sites, process of metal ions sorption onto sorbents happens very fast at the beginning of the process. However, during the

sorption process the number of available sites decreases together with metal concentration, decreasing lead removal rate. Electrostatic repulsion between already bonded cations can happen, too [17]. There are noticeable differences between sorbents affinity toward lead ions and in case of AP sorption process is more intensive: after 10 min the value of AP sorption capacity is almost 85% of equilibrium value while in the same moment value of BR sorption capacity is 73%.

The effect of sorbent concentration was studied and the results are presented at Figure 3c. It is known that the sorption capacity will be affected by variation of solid/liquid phase ratio. When initial lead concentration is fixed as in this case (200 mg/L), initial concentration gradient between sorbent and sorbate gets decreased with increasing amount of sorbent, which causes decrease of sorption capacity [18]. In this regard the solid/liquid ratio of 2 g/L was chosen.

In the Figure 3d, where effect of initial metal concentration onto sorption capacity was presented, it is evident that increase of initial metal concentration leads to higher sorption capacity. The enhancement of initial solution concentration is a mighty driven force to overcome mass transfer resistance of lead ions between solid and liquid phase [14]. The sorbent BR has a higher sorption capacity than AP: at initial lead concentration of 200 mg/L sorption capacity of BR is almost 60%, higher in comparison to AP.

The results of sorption experiments set on different temperature are presented in Figure S1. The lead adsorption capacity was reduced as the system temperature increased in the case of BR, dropping from 55.0 mg g$^{-1}$ (293 K) to 37.5 mg g$^{-1}$ (23 K), while in the case of AP it has an opposite trend: increase from 28.0 mg g$^{-1}$ (293 K) to 38.5 mg g$^{-1}$ (323 K).

The results of previous experiments enabled determination of optimal process parameters: pH 5.0, contact time 120 min, sorbent concentration 2 g/L and initial Pb$^{2+}$ con-centration 200 mg/L and 293 K.

*3.5. Kinetic Study*

Kinetic study gives information about reaction pathways and the mechanism of the sorption process [36]. To get insight of sorption kinetics of Pb$^{2+}$ onto AP and BR, pseudo-first order, pseudo-second order, together with Weber-Morris diffusion model were applied to obtained experimental date. Kinetic plots of these models are presented at Figures S2 and S3, while kinetic parameters are presented in Table 3.

According to the rate of correlation coefficients ($R^2 = 0.999$) pseudo–second order model is more suitable for prediction of the experimental data for both sorbents, indicating that the complexation and ion exchange mechanism are involved in lead sorption [19]. However, the pseudo-first order model shows also very high rate of its correlation coefficients (0.979 and 0.970 for AP and BR, respectively) indicating that this model is also suitable at first experimental points, where rapid sorption of lead occurs. The similar behavior has been observed earlier by Lopičić et al. [25] who investigated sorption kinetics of Cu$^{2+}$ onto peach stones, where authors notice that pseudo-firs order model is very applicable to kinetic date in initial rapid phase. According to the sorption capacity (qt) it is evident that sorbent BR has higher affinity toward lead ions in comparison to AP (50.7 and 34.7 mg/g, respectively). However, the applicability of the sorbent is not only determined by its capacity but it is affected also by the rapid kinetics. In order to ensure process effectiveness kinetics has significant practical importance. The sorption performance also can be described by calculating the half-life of sorption process (t$^{1/2}$) and this parameter describes required time for the drop of sorbate concentration by one-half of its initial value. From the obtained results (Table 3) it is clear that AP approaches equilibrium faster than BR. In just 1 min sorbent AP will drop initial metal concentration by half, which is twice as fast as sorbent BR.

**Table 3.** Kinetic parameters and correlation coefficients.

|  | AP | BR |
|---|---|---|
| $Ci$ (mg/L) | 200 | 200 |
| *Pseudo-first order model* | | |
| $q_t$ (mg/g) | 37.74 | 51.41 |
| $k_1$ (1/min) | 3.36 | 3.45 |
| $R^2$ | 0.979 | 0.970 |
| $\chi^2$ | 0.030 | 0.003 |
| *Pseudo-second order model* | | |
| $q_t$ (mg/g) | 34.70 | 50.74 |
| $k_2$ (g/mg/min) | 0.026 | 0.009 |
| $k_2 q_t$ (1/min) | 0.902 | 0.457 |
| $t_{1/2}$ (min) | 1.11 | 2.19 |
| $R^2$ | 0.999 | 0.999 |
| $\chi^2$ | 0.030 | 0.032 |
| *Weber-Morris diffusion model* | | |
| $K_{id1}$ (mg/(min$^{1/2}$ g)) | 9.47 | 12.30 |
| $C$ (mg/g) | 0.267 | 0.323 |
| $R^2$ | 0.992 | 0.993 |
| $K_{id2}$ (mg/(min$^{1/2}$ g)) | 0.878 | 4.18 |
| $C$ (mg/g) | 26.33 | 27.58 |
| $R^2$ | 0.979 | 0.970 |
| $K_{id3}$ (mg/(min$^{1/2}$ g)) | 0.148 | 0.181 |
| $C$ (mg/g) | 30.15 | 48.19 |
| $R^2$ | 0.836 | 0.433 |

In order to define if the surface characteristics of sorbents have effect on intraparticle diffusion, the diffusion of lead ions into the internal surface was studied. The Weber-Morris model is based on following assumptions: diffusion of lead ions to the external surface of the sorbent followed by intraparticle diffusion in to the pores of the sorbent and phase of equilibrium due to low residual lead concentration and deficit of active sites [20]. Those two processes overlap simultaneously. Since the plot passes through the origin, (Figure S3) the intraparticle diffusion is the only rate controlling step [25]. The diffusion rate ($K_{id}$) of lead ions onto both sorbents decreases gradually through stages, which is result of increased boundary layer thickness.

*3.6. Isotherm Studies*

The isotherm equilibrium studies were done in order to understand the nature of interactions between sorbents and sorbate and to determine maximum sorption capacity of both sorbents toward lead ions. The parameters obtained from isotherm models will provide some insight of bonding mechanism. Experimental date was fitted with two-parameter isotherm models (Langmuir and Freundlich) and with three-parameter models (Redlich-Peterson and Sips). All models equations are summarized in Table S1.

Langmuir isotherm model [37] is based on few assumptions: sorbent surface is energetically homogeneous, monolayer adsorption occurs and there are no interactions between sorbate species [18]. Equilibrium parameter $R_L$ is dimensionless constant from Langmuir isotherm model which expresses the type of isotherm: favorable ($0 < R_L < 1$), linear ($R_L = 1$), unfavorable ($R_L > 1$) and irreversible ($R_L = 0$). The Freundlich isotherm model [38] is applicable to the sorption process that happens onto heterogonous surface. When calculated parameter $1/n$ is less than 1 it is indication that adsorption happens at low concentration [39]. Sips [40] isotherm model is combination of Langmuir and Freundlich type of models. When initial metal concentration is low this model is much closer to Freundlich model, while at higher sorbate concentration it is closer to Langmuir isotherm model (also when factor $n_s$ is equal to 1) predicting monolayer adsorption. Redlich-Peterson model is also, compromise between Langmuir and Freundlich models [41] and at low sorbate concentration it follows the Langmuir isotherm model, while at high concentration this

model looks like Freundlich model. As value of Redlich-Peterson exponent (ß) is equal to 1 the Redlich-Peterson equation becomes a Langmuir equation. All isotherm models are graphically presented at Figure S4 while the calculated parameters are presented in Table 4.

**Table 4.** Isotherm parameters for $Pb^{2+}$ onto AP and BR.

| Modells | Parameters | AP | BR |
|---|---|---|---|
| Langmuir | $q_{max}$ (mg/g) | 32.09 | 72.99 |
| | $K_L$ (L/mg) | 0.166 | 0.030 |
| | $R_L$ | 0.015 | 0.075 |
| | $\chi^2$ | 0.096 | 8.71 |
| | $R^2$ | 0.995 | 0.975 |
| Freundlich | $Kf$ (mg/g)(L/mg)1/$n$ | 16.90 | 10.59 |
| | $1/n$ | 0.12 | 0.34 |
| | $\chi^2$ | 5.224 | 19.96 |
| | $R^2$ | 0.795 | 0.943 |
| Redlich-Peterson | $k_{RP}$ (L/g) | 4.861 | 2.505 |
| | $a_{RP}$ (L/mg) | 0.139 | 0.045 |
| | $\beta$ | 1.016 | 0.949 |
| | $\chi^2$ | 10.02 | 12.76 |
| | $R^2$ | 0.998 | 0.951 |
| Sips | $q_m$ (mg/g) | 31.68 | 79.77 |
| | $K_s$ (L/g) | 0.13 | 0.04 |
| | $n_s$ | 1.12 | 0.85 |
| | $\chi^2$ | 0.060 | 11.45 |
| | $R^2$ | 0.997 | 0.978 |
| | Best fit isotherm: AP: RP, S, L > > F BR: S, L > RP > F | | |

From the obtained experimental data both sorbents demonstrate significant potential for lead removal from aqueous solution. Among the applied models the three-parameter models (Redlich-Peterson and Sips isotherm models) provided better fitting to data than two-parameter models. In the case of AP, Freundlich correlation coefficient gives the poorest fit to data compared to other models. The correlation coefficients of other models applied (Sips, R-P and Langmuir) are very similar (0.997, 0.998 and 0.995, respectively), suggesting the monolayer adsorption. In case of BR the situation is different and correlation coefficient of all models are very similar suggesting that distribution of $Pb^{2+}$ between sorbent and sorbate is much more complex. The maximum sorption capacities from Sips model are 31.7 and 79.8 mg/g for AP and BR, respectively. Also, since parameters $n_s$ are close to 1, the Sips isotherm model becomes a Langmuir model and implies a homogenous adsorption process [42]. Rima et al. [43] study the heavy metals sorbent based on beetroot fibers for treatment contaminated water and concluded that this sorbent possess functional groups that are able to bind heavy metal cations, particularly C=N detected at 1640 cm$^{-1}$, which forms stable complexes with a large number of heavy metals. However, in this study, there is no information about sorption capacity. Chand and Pakade [21] who studied lead sorption onto dry AP and activated carbon prepared from AP, obtained maximum sorption capacities 16.4 and 16.0 mg/g, (from Langmuir isotherm model), indicating that in this study maximum sorption capacity is 100% higher. Since the particle size in their study was similar to those in our study, explanation could be found in different way of sample preparation. In this study the AP with water content about 75% [28] was dried by efficient low-cost method at industrial scale in dehydrator "Solaris". It is well known that among various techniques freeze drying process successfully preserve bioactive compounds. However, Zlatanović et al. [28] confirmed that drying using this dehydrator under specific conditions, not only valuable compounds remain successfully preserved but this method offers more environmental friendly approach (the drying time and energy consumption are much lower than in freeze drying). Authors strongly believe

that dehydrator operational features contributed to preservation of functional groups which plays important role in metals binding. According to Nawirska [22] the lead ion binding capacity of the various investigated food pomace components decrease in the following order: polyphenols >pectins >hemicelluloses >cellulose >lignin. AP sample that we have used in our experiments has 4.6–8.1 mg GAE/g of total phenols; 3.0–4.5 g/100 g dw of pectin; 15–20 g/100 g dw of cellulose [30].

*3.7. Thermodynamics Studies*

In Supplementary Materials, Figure S5, Van't Hoff plots obtained by plotting of $\ln K_e^0$ vs. $1/T$ for both sorbents are presented. As a result, Van't Hoff plots have different sign of the slopes, indicating different nature of sorption. In order to understand opposite trends in the sorption thermodynamics, the composition of the organic constituents of AP and BR sorbents play significant role. As it can be seen from Table 1, AP mainly consists of carbohydrates, insoluble fibers and cellulose, indicating the presence of the functional groups: –COO, –CO, –NH2, –CH2 and –OH, that are highly responsible for the metals binding. On the other hand, BR is composed of the same constituents, which are present in lesser amounts, while the amount of proteins (rich in –N–H and N=N groups) is much higher, 13.85% compared to 3.2% in AP. In the same time, later in the text FTIR analyses (Table 5) have confirmed presence of free and absorbed water with peaks more pronounced in AP than in BR sample, indicating that certain amount of energy must be used in this desorption, contributing to endothermic effect of lead sorption onto AP. According to Romero-González et al. [44], small positive metal–ligand heats ($\Delta H^0 < 16$ kJmol$^{-1}$) are characteristic of metal ions coordination with carboxyl groups, whereas coordination with amino and sulfhydryl groups results in large negative enthalpy changes [45]. Having in mind the difference in composition of the organic constituents of AP and BR, main functional groups identified by FTIR, it is expected that these two sorbents might have different sorption behavior in the sense of temperature influence.

**Table 5.** Thermodynamic parameters of $Pb^{2+}$ sorption by AP and BR sorbents.

| Sample | $\Delta H^0$ (KJ/mol) | $\Delta S^0$ (J/mol/K) | $\Delta G^0$ (KJ/mol) | | |
|--------|----------------------|------------------------|-----------------------|---------|---------|
| | | | **293 K** | **308 K** | **323 K** |
| AP | 19.10 | 108.25 | −13.19 | −13.68 | −14.40 |
| BR | −28.43 | −41.94 | −16.12 | −15.14 | −14.35 |

Summarized data of calculated thermodynamic parameters for both AP and BR are presented in Table 5.

As can be seen from Table 5, $\Delta G^0$ has negative values for both samples, indicating that $Pb^{2+}$ sorption onto both sorbents is spontaneous process; $\Delta G^0$ decreases with temperature increase in the case of AP, showing that the sorption is more favorable at higher temperatures; for the sorbent BR we have opposite case. Presented results of $\Delta S^0$ was positive in the case for sorption of $Pb^{2+}$ on AP, and negative on BR. The positive value of $\Delta S^0$ implied that $Pb^{2+}$ sorption on the sorbent AP increases the degree of freedom of the adsorbed species and increases the randomness at the solid/liquid interface [19], whereas the negative value of $\Delta S^0$ implies the reverse. Positive value of $\Delta H^\circ$ suggests that sorption of $Pb^{2+}$ onto AP is endothermic; while the negative value in case of BR indicates that the sorption process is exothermic. Generally, small enthalpy changes less than 84 KJ/mol indicate physical sorption [46] while the small positive metal–ligand heats ($\Delta H^0 < 16$ KJ/mol) are characteristic of metal ions coordination with carboxyl groups [44]. At the same time the heat evolved during physical adsorption is of the same order of magnitude as the heats of condensation, i.e., 2.1–20.9 KJ/mol [47]. Considering these, it might be estimated that mostly physical sorption is involved in $Pb^{2+}$ binding onto sorbents AP and BR.

### 3.8. Sorption Mechanism

3.8.1. Ion Exchange Mechanism

The involvement of ion exchange mechanism was investigated by following the release of exchangeable ions from sorbent surface after the process of sorption. The ratio between bonded $Pb^{2+}$ and released exchangeable ions ($Na^+$, $K^+$, $Ca^{2+}$, $Mg^{2+}$ and $H^+$) from sorbents at three different initial concentrations was calculated by following equation:

$$R_{\frac{b}{r}} = \frac{[Pb^{2+}]}{[Ca^{2+}] + [Mg^{2+}] + \frac{[Na^+]}{2} + \frac{[K^+]}{2} + \frac{[H^+]}{2}} \tag{7}$$

If the ion exchange mechanism was the only mechanism that occurs during the process of sorption the ratio $R_{b/r}$ would be equal to unity. However, in a case of AP the ratio between bonded lead ions and released cations is higher than 1.0: at initial concentration of 40, 120 and 210 mg/L the $R_{b/r}$ are 1.3; 1.2 and 1.3, respectively. These results are graphically presented at Figure 4. This is confirmation that ion exchange mechanism is involved in sorption mechanism of lead ions onto AP, but it is not the only bonding mechanism. From the Figure 4 it can be seen that sodium ions are not involved in the ion exchange mechanism of the AP, but all other ions are involved almost equally.

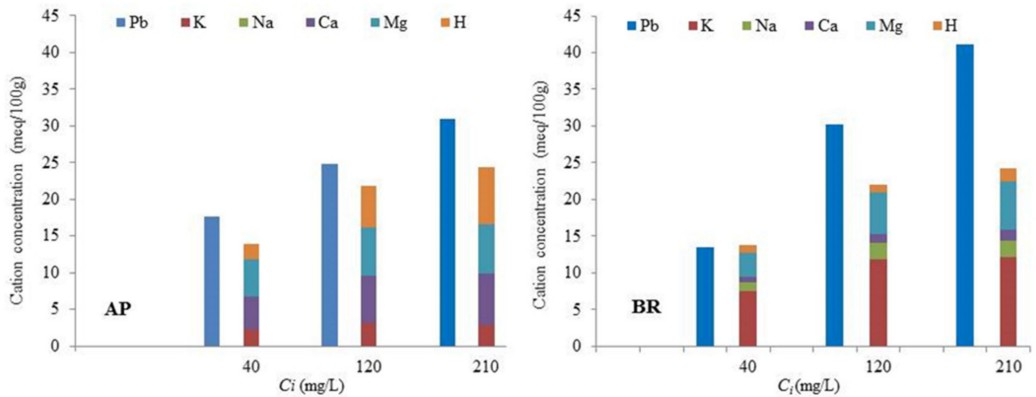

**Figure 4.** Ratio between bonded lead ions and released cations from AP and BR during sorption at different initial concentration (operation parameters were pH 5.0, contact time 120 min, sorbent concentration 2 g/L and initial $Pb^{2+}$ concentration 40, 120 and 210 mg/L).

At initial concentration of 40 mg/L in the BR ion exchange mechanism is the only mechanism involved in sorption process, which was also confirmed by calculated values of $R_{b/r}$ which are close to unity (0.98). But at higher initial concentration calculated value of $R_{b/r}$ is far more than 1 (1.4 and 1.6) which means that at higher initial concentration some other mechanism is involved beside ion exchange mechanism. It is noticeable that dominant ion in the exchangeable position in the BR is potassium, followed by magnesium ions (in accordance with the results of CEC—Table 2). Replacement of the alkali and earth alkali ions by metal indicates ionic binding while replacement of protons by metal ions indicates covalent binding [42].

3.8.2. ATR-FTIR of AP and BR before and after Sorption of Lead

In order to observe changes in functional groups before and after lead sorption onto AP and BR ATR-FTIR analysis were performed. The spectra of the AP and BR before and after sorption of lead ions are presented at Figure 5. The dominant bands in FTIR spectra of the AP were labelled elsewhere [29]. The spectrum of AP before and after sorption revealed that the most of the band's intensity decreased after sorption of lead ions. Also, the position of band corresponds to –OH stretching vibration has been changed from 3328 cm$^{-1}$ to 3334 cm$^{-1}$, indicating a possible complexation process [19]. The similar results were observed when Chand and Pakade [21] analyzed FTIR spectrum of the apple pomace before

and after sorption of lead ions. According to them –OH, –C–H, –CO, –C=O and –C–C groups are responsible for lead ions sorption which is in accordance with our findings.

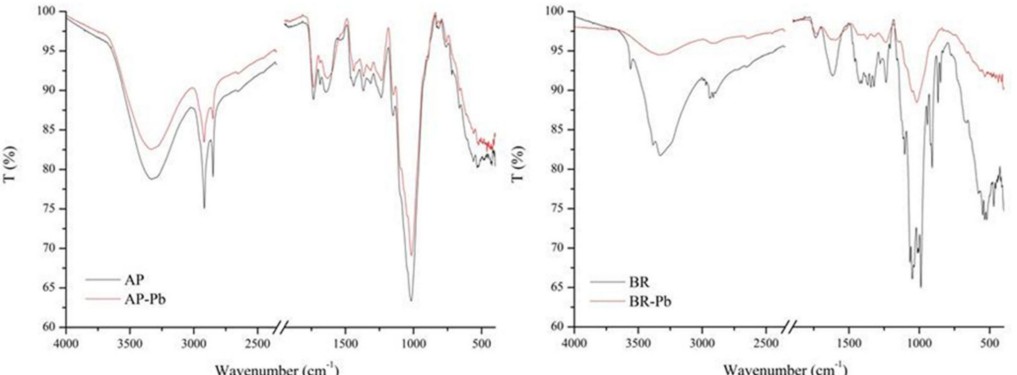

**Figure 5.** ATR-FTIR spectra of AP and BR before and after sorption of lead ions.

ATR-FTIR spectrum of BR shows band at 3560 cm$^{-1}$ assigned to N–H stretching vibration of the amide [48]. The wide band observed at wavenumber between 3500–3000 cm$^{-1}$ corresponds to –OH stretching vibration of water and phenols –OH groups, while the bands between 2950–2800 cm$^{-1}$ correspond to –CH and –CH$_2$ groups [29]. The band at 1735 cm$^{-1}$ corresponds to aliphatic aldehyde –C=O stretching [49]. The band detected at 1640 cm$^{-1}$ assigned to C=N [43]. According to Pradhan et al. [48] bands between 1459–1408 cm$^{-1}$ correspond to amide group, while bands between 1385–1322 cm$^{-1}$ correspond to –N=N stretching vibration. The bands at 1278 and 1237 cm$^{-1}$ correspond to –C–O–C stretching from hemicellulose and phenols [49]. The band between 1065–989 cm$^{-1}$ originate from –C–C, –C–OH, –C–H group vibrations from cellulose and phenols [29].

The intensity of all peaks after lead sorption by both sorbents noticeable decreased. This change is especially visible in ATR-FTIR spectrum of the BR. After lead sorption intensity of the peak corresponding to –OH stretching vibration drastically decreased suggesting the possible complex formation between –OH groups and lead [19,50]. Since, intensity of the peaks in fingerprint region also decreased and the wavenumber shifted indicating involvement of amide, –C–C, –C–OH, –CH groups in lead adsorption, it can be concluded that the complexation process is involved in the adsorption (which was confirmed by kinetic study).

### 3.8.3. Zeta Potential of AP and BR before and after Sorption of Lead

Significant factor affecting process of sorption is charge of sorbent surface. This is a result of presence of various acidic groups (carboxyl, phosphate, amino, hydroxyl, sulfhydryl groups) responsible for binding of metal ions [51]. To investigate electrical potential of sorbents particles surface before and after lead sorption, zeta potential was measured and results are given in Table 6. Since, all experiments were done at pH 5, the water as a medium for determination of zeta potential was adjusted at this value (it also corresponds to the pH value with maximum sorption capacity).

**Table 6.** Zeta potential of AP and BR before and after sorption of lead.

| Sample | Zeta Potential (mV) | pH Value * |
|---|---|---|
| AP | −36.8 | 5.00 |
| AP-Pb | −33.8 | 3.79 |
| BR | −40.1 | 5.00 |
| BR-Pb | −24.4 | 4.37 |

* Initial and final pH value of the sorbate after sorption process.

From Table 6 it is evident that both sorbents at this particular pH value (pH = 5.0) possess negative zeta potential, indicating the presence of compounds which contains negatively charged functional groups on its surface such as carboxyl, carbonyl and hydroxyl groups, which was also proven by FTIR analyses. Since negative value of the zeta potential revealed that the surface charges of both sorbents are negative, it can be concluded that lead sorption occurred through electrostatic interaction which is in accordance with ion exchange mechanism study.

From results in Table 6 it can be observed that when lead cations are adsorbed onto sorbent surface, zeta potential is changed. In case of the AP slight change of zeta potential before and after sorption can be observed (from −36.8 to −33.8 mV, respectively). However a significant drop of the pH value after sorption can be observed as well, indicating major role of hydrogen ions in bonding mechanism. This is in accordance with observed results from ion exchange mechanism analysis, where it was observed that the hydrogen ions are more exchanged along with the rise of metal solution concentration. In case of the BR zeta potential before and after sorption has been changed from −40.1 to −24.4 mV, together with slight drop of pH value from 5.00 to 4.37. This is in accordance with obtained results from ion exchange mechanism analysis, where it was observed (from Figure 4) that hydrogen ions have negligible role in ion exchange mechanism of BR in comparison to AP.

## 4. Conclusions

This study was focused on sorption performance of dried apple and beetroot pomace toward lead ions from aqueous solution. Both materials showed excellent sorption abilities and experimental data obtained by using the AP in sorption experiments were well-fitted to Redlich-Peterson, Sips and Langmuir isotherm models, indicating monolayer sorption. In case of the BR the situation was different and correlation coefficients of all models were very similar suggesting that sorption mechanism is much more complex. The maximum sorption capacities from Sips are 31.7 and 79.8 mg/g for AP and BR, respectively.

Cation exchange capacity (CEC) for AP and BR is 24.75 and 95.85 meq/100 g, respectively. BR has significantly higher CEC, due to the presence of potassium ions which are dominant ions in the exchangeable position. The study of sorption mechanism confirmed that ion exchange mechanism plays an important role during the process of sorption, and in case of the AP it was observed that the hydrogen ions are more exchanged with lead ions along with the rise of metal solution concentration.

The obtained maximum sorption capacity of AP is two times higher in comparison to maximum sorption capacity obtained by other researchers. The way of sample preparation could contribute to preservation of functional groups which have major role in metal binding.

**Supplementary Materials:** The following supporting information can be downloaded at: https://www.mdpi.com/article/10.3390/pr11051343/s1, Figure S1: Effect of temperature on sorption capacity; Figure S2: Pseudo-first (left) and pseudo-second order (right) kinetics plots of Pb2+ onto AP and BR; Figure S3: Weber-Morris diffusion plots of Pb2+ onto AP and BR; Figure S4: Isotherm models of Pb2+ sorption onto AP and BR; Figure S5: Van't Hoff plots for the adsorption of Pb2+ onto AP and BR; Table S1: Models used for evaluation of lead sorption onto AP and BR.

**Author Contributions:** Conceptualization, T.Š. and M.S.; methodology, T.Š. M.S. and Z.L.; validation, T.Š. and Z.L.; formal analysis, T.Š. and Z.L.; investigation, T.Š. and M.S.; resources, S.G. and S.Z.; data curation, A.A.; writing—original draft preparation, T.Š.; writing—review and editing, Z.L., S.G., F.P. and S.Z.; visualization, F.P.; supervision, T.Š., Z.L. and S.G.; funding acquisition, T.Š. M.S. and Z.L. All authors have read and agreed to the published version of the manuscript.

**Funding:** This research was funded by the Ministry of Science, Technology Development and Innovation of the Republic of Serbia, grant numbers: 451-03-47/2023-01/200023, 451-03-47/2023-01/200051 and 451-03-47/2023-01/200168.

**Institutional Review Board Statement:** Not applicable.

**Informed Consent Statement:** Not applicable.

**Data Availability Statement:** Not applicable.

**Acknowledgments:** The author would like to thank to juice producer Fruvita, Smederevo and Healthy Organic-Selenča, Serbia for samples supply.

**Conflicts of Interest:** The authors declare no conflict of interest.

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
