# Peer review of "Food Waste (Beetroot and Apple Pomace) as Sorbent for Lead from Aqueous Solutions—Alternative to Landfill Disposal"

_processes, doi:10.3390/pr11051343_

Round 1

Reviewer 1 Report

After reading this manuscript the following shortcomings were disclosed.

1). Remark on English. The entire article contains many grammatical and stylistic errors and therefore needs to be carefully edited.

For example, Abstract. There are so many grammatical errors in the abstract that it needs to be rewritten, e.g., as follows:

This article presents studies, whose main goal was to minimize food waste. To achieve this goal, it is necessary to expand the scope of their application, for example, for the purification of polluted water from heavy metals. Millions of tons of waste from the fruit and vegetable industry, including pomace of apples and beetroots, are thrown into landfills, posing a danger to the environment. In order to solve the problems with the disposal of these wastes, the authors investigated their sorption potential for the removal of lead from wastewater. The sorbents, dried apple (AP), and beetroots (BR) pomaces were characterized by various methods (study of composition, zeta potential, FTIR-ATR, and SEM-EDX). Various models of sorption kinetics and sorption isotherms were analyzed. Kinetical studies under optimal conditions showed that the sorption process occurs through complexation and ion exchange and the determining stage limiting the rate of sorption is the diffusion of lead ions in the sorbent. The maximum sorption capacity was 31.8 and 79.8 mg/g for AP and BR, respectively. The thermodynamic data revealed the spontaneous sorption of lead ions by sorbents. The temperature rise contributes to the sorption increase by the AP sorbent, while for the BR sorbent, the opposite effect is observed. The obtained results showed that apple and beetroots pomaces can serve as effective renewable materials for the preparation of sorbents, contributing to the solution of complex environmental problems.

2). Remark: It is not clear why the authors chose the sorption of lead ions rather than the ions of another heavy metal from wastewater. Justification is required, which wastewater (which industries) is polluted exclusively with lead ions and therefore needs to be cleaned from ions of this heavy metal.

3). Regarding 2.4 Sorption experiments.

Remark 1 on Eq. (4). It needs to explain what is ce? Is it the same as cf in eq. (1) or another?

Remark 2: Why the initial concentration of Pb cations 200 mg/L was chosen? Justification is required.

Remark 3: Why concentration of sorbent was expressed in g/50 ml instead of g/100ml or percent?

Lines 161-162. ...sorption onto both sorbents was conducted in the range from 20 to 50°C. Remark 4: Since thermodynamical calculation uses temperature in K, the temperature in oC should be replaced with K, i.e., in the range from 293 to 323 K (see lines 206-207).

4). Regarding 3.1. Characterization of sorbents.

Remark 1: Organic composition of AP. Cellulose and pectin are different polysaccharides and therefore instead of their sum, the content of each of these components should be indicated.

Remark 2: Organic composition of BR. The detailed organic composition of BR must be specified. It is not enough to be limited to a common phrase that its “composition was characterized as well”.

5). Regarding 3.2. SEM-EDX.

Lines 231-232. As can be seen, both samples are layered with noticeable gaps between, while BR has... Remark 1: Fig. 1 (compare a & c) shows that only BR has layered morphology.  Thus, this sentence should be corrected as follows, “As can be seen, the BR sample has a layered fibrous morphology with noticeable gaps between fibres and pores inside them”

Remark 2: regarding EDX. This method is different from SEM; therefore, another (separate) Figure is required for EDX images. Additional remark: the presented EDX images are low-quality, and they must be improved.

6). Regarding 3.4. Effect of operation parameters on sorption process.

Figure 2. Remark 1:  The red-violet and dark-blue lines must be explained, which refers to AP and with to BR. Additional remark: Why sorbent concentration is expressed in g/50 ml (c), and in the caption (a, b, d) this concentration is expressed in g/L (2 g/L)? A unified unit of measure is required to use.

Remark 2: Figure 2, c. It shows a decrease in sorption value with increasing concentration of sorbent. Theoretically, with properly selected concentrations of ions, the volume of the solution, and the concentration of the sorbent, the specific sorption per unit mass of the sorbent (q, mg/g) is not dependent on sorbent concentration. If the adsorption value q decreases, this means that the sorption conditions were not optimal.

Remark 3: Figure 2, d. The authors wrote that the initial concentration of Pb cations was 200 mg/L. Why in Figure 2, d, the initial concentration is >300 mg/L?

Lines 308-309. The temperature should be expressed in K.

7). Regarding 3.4. Isotherm studies

Remark: Lines 400-404. In this section, the authors describe the effect of organic constituents of AP on the binding ability of Pb ions.  The same analysis is required for BR sorbent; for this purpose, the authors should indicate the composition of organic constituents also in the BR sample.

8). Regarding 3.5. Thermodynamics studies

Remark: The study of the thermodynamics of the sorption process gave strange and unexpected results. For the AP sorbent, the enthalpy, and entropy are positive, which is typical for physical surface adsorption, while for the BR sorbent, these thermodynamic characteristics are negative, which is typical for chemisorption. This also corresponds to different temperature dependences of sorption for these sorbents. In this regard, it is imperative to study the composition of the organic constituents of these sorbents in order to understand why there are such cardinal differences in the mechanism of sorption of sorbents based on fruit (AP) and vegetable (BR).

In this regard, it cannot agree with the conclusion of the authors that “it can be assumed that physical sorption is mainly involved in the binding of Pb2+ on AP and BR sorbents” (lines 428-429).

9). Regarding 3.6.1 Ion exchange mechanism

Remark: Given the cardinal differences in the thermodynamic characteristics and temperature dependences of the sorption of Pb ions by the two sorbents AP and BR, it cannot be agreed that the ion exchange mechanism is the only sorption mechanism for both sorbents. In addition, ATR-FTIR studies also showed that the functional groups of these sorbents are involved in the sorption process.

Author Response

Answers to Reviewers comments

First of all, we would like to thank Reviewers for their time and comments which will improve the quality of our paper. We have included all the responses and comments in revised form of our paper by using “track changes”. English language corrections were done by native English speaker. We hope that we have fulfilled reviewers demand, and that improved version of the paper satisfies demands to be published in Your Journal. Point-by-point responses to the reviewers' comments (the reviewers’ comments are bold and in italics) are given below.

Please see the attchment.

Reviewer 2 Report

In this work, Šoštarić et al. have investigated the potential usage of food waste (beetroot and apple pomace) as sorbent to remove heavy metal lead from aqueous solution. The effect of various experimental parameters, including pH, temperature, contact time, sorbent loading, and lead concentration on lead sorption are investigated. Their provided sorption mechanism suggest ion exchange mechanism plays an important role. Overall, the results demonstrate that beetroot and apple pomace could serve as an efficient renewable sorbent to remove lead. This work is of interest to the audience of Processes in general. There are few minor concerns, though, that need to be addressed before acceptance.

1.     The authors have cited previous publications on using food waste to remove heavy metals such as Cu and Pd from aqueous solution (ref 14, 16-17, 24-30). However, these works were not mentioned in the introduction. The authors are recommended to include one more paragraph in the introduction to summarize the previous efforts on this field.

2.     Figure 1. I would suggest the authors to reorganize the panels, making the left-side panels represent AP and the right-side panels represent BR, therefore they can be compared side-by-side.

3.     Table 2: How the standard deviations in ∑ are calculated? These numbers seem not accurate. Additionally, are the CEC values of 6.1 and 22.1 standing for lead? This should be clarified in the Table.

4.     Ln 251: change the significant figures of the CEC values to be consistent with the numbers in Table 2.

5.     Figure 2C: Please use the same unit for the x-axis (g/50ml) and the figure caption (g/L). Can the authors also plot the total amount of absorbed lead vs sorbent loading and put that figure in SI?

Author Response

Answers to Reviewers comments

First of all, we would like to thank Reviewers for their time and comments which will improve the quality of our paper. We have included all the responses and comments in revised form of our paper by using “track changes”. English language corrections were done by native English speaker. We hope that we have fulfilled reviewers demand, and that improved version of the paper satisfies demands to be published in Your Journal. Point-by-point responses to the reviewers' comments (the reviewers’ comments are bold and in italics) are given below.

Round 2

Reviewer 1 Report

Eq. (1). Remark: The authors used the new symbol Ce instead of Cf in the note. But in equation (1) the old symbol, Cf, remains, which must be replaced with a new symbol, Ce.

Besides, English requires additional editing.

Author Response

Dear Reviewer,

Thank you for your helpful comments. 

Eq. (1). Remark: The authors used the new symbol Ce instead of Cf in the note. But in equation (1) the old symbol, Cf, remains, which must be replaced with a new symbol, Ce.

 Response: The comment has been accepted and the equation (1) has been revised.

Besides, English requires additional editing.

Response: English language corrections were done by native English speaker.